# Can Chlorophyll a Fluorescence and Photobleaching Be a Stress Signal under Abiotic Stress in *Vigna unguiculata* L.?

**Marcelo F. Pompelli** [1,*,†] , **Daniela Vegliante Arrieta** [2,†] , **Yirlis Yadeth Pineda Rodríguez** [2,†] ,
**Ana Melisa Jiménez Ramírez** [2,†] , **Ana Milena Vasquez Bettin** [2,†] , **María Angélica Quiñones Avilez** [2,†] ,
**Jesús Adolfo Ayala Cárcamo** [2,†] , **Samuel Giovanny Garcia-Castaño** [2,†] , **Lina María Mestra González** [2,†] ,
**Elias David Florez Cordero** [2,†] , **Marvin José Perneth Montaño** [2,†] , **Cristian Camilo Pacheco Mendoza** [2,†] ,
**Anthony Ricardo Ariza González** [2,†] , **Alberto José Tello Coley** [2,†] , **Alfredo Jarma-Orozco** [1,†]
**and Luis Alfonso Rodriguez Paez** [1,†]

1 Facultad de Ciencias Agronómicas, Universidad de Córdoba, Montería 230002, Colombia
2 Post-Graduation in Crop Science, Universidad de Córdoba, Montería 230002, Colombia
\* Correspondence: marcelo@fca.edu.co
† These authors contributed equally to this work.

**Abstract:** Greenhouse gas emissions continue raising the planet's temperature by 1.5 °C since the industrial age, while the world population growth rate is 1.1%. So, studies aimed at food security and better land use are welcomed. In this paradigm, we choose *Vigna unguiculata* to test how it would behave in the face of severe abiotic stresses, such as drought and salt stress. This study shows that under abiotic stresses *V. unguiculata* tries to overcome the stress by emitting chlorophyll a fluorescence and promoting photobleaching. Thus, fewer photons are directed to photosystem I, to generate lethal reactive oxygen species. The antioxidant system showed a high activity in plants submitted to drought stress but fell in salt-stressed plants. Thus, the reductor power not dissipated by fluorescence or heat was captured and converted into hydrogen peroxide ($H_2O_2$) which was 2.2-fold higher in salt-stressed *V. unguiculata* plants. Consequently, the malondialdehyde (MDA) increased in all treatment. Compiling all data, we can argue that the rapid extinguishing of chlorophyll a fluorescence, mainly in non-photochemical quenching and heat can be an indicator of stress as a first defense system, while the $H_2O_2$ and MDA accumulation would be considered biochemical signals for plant defenses or plant injuries.

**Keywords:** osmotic stress; drought stress; waterlogging; chlorophyll a fluorescence; antioxidative enzymes; food security; global climate change

## 1. Introduction

The planet Earth is on the verge of a total collapse. Since the industrial age, the global temperature has increased by about 1.5 °C [1], due to the increase in the emission of greenhouse gases, such as $CO_2$, CO, $N_2O$, and $NH_4^+$ [2]. The world's population is expected to increase in 2100 to 11 billion inhabitants [3,4]. With a higher world population there will also certainly be a greater demand for food. Global climate change can make this scenario worse by turning areas that are currently prime areas for planting into degraded areas and areas that are currently degraded into areas which poorly support crop production, exposing cultivated plants to conditions that aren't experienced naturally [4–7]. Increase in sea-level rise, flooding, and salinity stresses are potentially damaging to global food production [8].

It is known that drought is one of the major abiotic stresses causing negative impacts to crops [9]. Salinity is another major issue affecting photosynthesis and crop production worldwide. It is estimated that approximately 20% of land areas and 33% of irrigated crop lands suffer from high salinity. It is also estimated that each year 10% more irrigated areas

become salinized [10–13] and this trend is pervasive throughout the globe, reaching ~830 to 950 million hectares [13]. Saline stress is specifically known to cause disturbances in photosynthesis since salinity increases soil osmotic potential, reduces water uptake by roots, and leads to a decrease in plant growth [14]. Both drought and salinity stresses seriously affect plant growth and result in serious restrictions on agricultural crop production.

Currently, more than 800 million people around the world go to bed hungry every night, most of them small farmers who depend on agriculture to earn a living and feed their families. Despite an explosion in urban slum growth over the past decade, nearly 75% of poor people in developing countries live in rural areas [15]. Thus, we can connect the future global food demand to the role of agricultural and food science in producing and stabilizing foods to meet this global food demand [16]. Enhancing the resilience of agricultural production to climate change requires promising alternatives to current practices. It is also necessary to cultivate crops that, under stress conditions of low water and high salinity, maintain a high potential and a productive quality. The main goal of this paper is to evaluate the position played by the cowpea in the face of drought stress due to a lack of irrigation, waterlogging and osmotic stress. Here, we discuss the role of chlorophyll a fluorescence, photobleaching, and how the antioxidant system behaves under stress.

Therefore, effective solutions are needed to deal with a decrease in the amount of food produced as a result of climate change, especially in the case of salt sensitive or moderately sensitive crops like the cowpea. The cowpea (*Vigna unguicuata* L. Walp) belongs to the Fabaceae family and is consumed as a green pods or dried seeds. It is an inexpensive food, yet it is an important source of protein and some essential amino acids, like tryptophan and lysine, vitamins, and minerals. In 2020 it's cultivation area was 15 million hectare with production of 9 million ton of dry beans [17], with Africa producing nearly 7.1 million. Nigeria, the largest producer and consumer, accounts for 48% of the production in Africa and 46% worldwide [16]. This species is used in sustainable agriculture because it is relatively tolerant to drought [18], high temperatures and high salinity, in addition to having the ability to fix atmospheric nitrogen and reduce soil erosion; therefore, it could grow better in arid and semi-arid areas and under stress conditions than other crops [19]. However, the degree of tolerance to stressors is usually cultivar or variety dependent [18,20–23]. Responses such as rapid stomatal closure is one of the strategies to avoid water stress. With stomatal closure, less carbon is taken up by the plant and therefore, less carbon will be reduced through the Calvin-Benson cycle. As is known, this cycle is described as a cycle of the reduction of carbon to sugars and for this, $CO_2$ (coming from the stomata of NADPH and ATP that comes from the ETR is used. Therefore, in the absence of carbon reduction, both NADPH and ATP remain in their reduced form, which generates a cascade of reduced molecules upstream that resulted in the unconditional donation of these electrons to oxygen, thus forming ROS, or in the cycle of photochemistry that is not sustainable over time and which also contributes to the formation of ROS In general, there are two ways to avoid or dissipate this loss of element synthesis capability. The reduction of chlorophyll synthesis or its degradation by chlorophyllases [24], is a mechanism that we will call photobleaching here. The second and faster way to avoid the overproduction of ROS is the dissipation of the chemical reaction in a safe way, using the fluorescence emission of chlorophyll a [25,26], which in general dissipates this in the form of heat or in the production of molecules, which in their synthesis use a chemical reaction, thus alleviating the overproduction of ROS [27]. However, there is a lack of holistic studies that comprehensively explain the tolerance to abiotic stress of cowpeas, so this study sought to determine the biochemical, physiological, and molecular mechanisms of the reduction of the gas exchange caused by abiotic stress.

## 2. Materials and Methods

### 2.1. Study Area and Plant Material

The experiment was conducted at the Faculty of Agricultural Sciences, University of Córdoba, Colombia (8°47′3″ N and 75°51′51″ W; 14 m a.s.l), under greenhouse conditions. The average annual rainfall of 1346 mm, with a relative humidity of 85%, an average annual

temperature of 27.4 °C, and an annual sunshine of 2108 h [28], that it is condensed by Martonne aridity index [29] of 40.6 [30].

The commercial variety Caupicor 50 of cowpea beans (*Vigna unguiculata* L. Walp) with seeds of excellent phytosanitary quality with germination percentage greater than 95% were used for sowing. A hundred seeds were soaked to germinate in polypropylene trails $40 \times 30 \times 10$ cm filled with substrate composed of a mixture of alluvium and loamy-silty texture, in proportion (1:1). 15 days after soaked, the 15 cm-seedling were transplanted to plastic bags of 1.5 L capacity, where all plants were maintained to acclimatize to a new environments by 15 days. After then, plants with two true leaf pairs were used in this experiments. The 30-days-plants were subjected to waterlogging for $\frac{3}{4}$ of the bag under water surface, to drought, by total withdrawal irrigation, and two osmotic potential ($\Psi$s = $-0.44$ MPa and $\Psi$s= $-0.89$ MPa) provided by NaCl, plus control. The experiment started on 19 March 2022 and was carried out until the plant showed signs of stress or in the 12th day after the onset of stress.

## 2.2. Leaf Gas Exchange Parameters and Chlorophyll a Fluorescence

The parameters of gas exchange and chlorophyll a fluorescence started to be measured on the 1st day of applying stress (day zero) and after this, every two days. The leaf gas exchange and chlorophyll a fluorescence utilized the 2nd attached fully expanded leaf from the apex to determine stress, using a portable open-flow infrared gas analyzer (LI-6400XT; LI-COR Inc., Lincoln, NE, USA) with integrated fluorescence chamber heads (LI-6400-40; LI-COR Inc.) as described in details on Pompelli, et al. [31].

## 2.3. Biochemical Analysis

A fully expanded leaf was harvested, soon after the last gas exchange and fluorescence measurements. Approximately 50 mg of leaf sample was used for the Chl and Car extraction using microtubes containing 2 mL of dimethyl sulfoxide (DMSO). The microtubes were involved in aluminum foil and taken to thermo-shaker (Multitherm, Benchmark Scientific, Sayreville, NJ, USA) incubated at 40 °C by 30 h at 200 rpm. Following this, 200 mL were transferred to microvials and evaluated in a microvial reader (ThermoScientific™ Multiskan™ GO, Missouri City, TX, USA).

Cellular damage was analyzed through (*i*) malondialdehyde (MDA) accumulation, estimated as the content of total 2-thiobarbituric acid-reactive; and (*ii*) $H_2O_2$ accumulation, estimated via the ferrous oxidation-xylenol orange method [32]. To determine free proline, the samples were ground in a 3% solution of 5-Sulfosalicylic acid and estimated as previously described [33].

## 2.4. Assaying the Activity of Antioxidant Enzymes

Approximately 50 mg of leaf sample was ground in 2 mL of 0.3 M potassium phosphate buffer (pH 7.8) solution with an added 1% of polyvinylpolypyrrolidone. The microtubes were mixed for 30 s and then spun in a centrifuge in a 4 °C, $15.000 \times g$, for 15 min. The supernatant phase was transferred to a new microtube and used for measurements of antioxidative enzymes, while the pellet was used to determine soluble proteins [34]. The activity of glutamine synthetase (GS; EC 6.3.1.2) was measured by the production of g-glutamyl-hydroxamate at 530 nm. Key antioxidant enzymes, including ascorbate peroxidase (APX; EC 1.11.1.11) and catalase (CAT; EC 1.11.1.6), were estimated by measuring the rate of decomposition of $H_2O_2$ at 290 nm and 240 nm, respectively. Superoxide dismutase (SOD; EC 1.15.1.1) was estimated by measuring its ability to inhibit the photochemical reduction of *p*-nitro-blue-tetrazolium chloride at 560 nm. Further details have been documented previously [32].

## 2.5. RT-PCR

For RT-PCR analysis, 1 g leaf samples were collected from five replicates of each treatment, following 10 days of stress exposure, and immediately ground to a fine powder

in the presence of liquid Nitrogen with subsequent storage at $-80\,^\circ$C until used. Total RNA extraction was achieved with the NucleoSpin RNA Plant kit (Macherey Nagel, Düren, Germany), RNA integrity and DNA contamination were evaluated on a 1% agarose gel, RNA concentration and quality were determined on a NanoDrop™ One/OneC spectrophotometer (Thermo Scientific). First strand cDNA was prepared using the High Capacity cDNA Reverse Transcription Kit (Applied Biosystems, Foster City, CA, USA) and a total RNA concentration of 1000 mg/mL, according to the manufacturer's protocol. The resulting cDNA mix was diluted 1:10 and stored at $-20\,^\circ$C. For gene expression analysis, a total of 13 genes related to abiotic stress were studied with the protocol pre-established by [27].

### 2.6. Total Biomass and Leaf Area Estimation

In the final portion of the experiments, length and width of the all leaves were measured for an estimated total leaf area [35]. Afterwards, all leaves, shoots, and roots were individually placed in paper bags and the dry weight was recorded after the material was oven-dried at 75 $^\circ$C for 72 h.

### 2.7. Experimental Design and Statistical Analyses

The experiments were conducted in a completely randomized block design with four treatments (waterlogging, drought, and two different concentrations of NaCl) plus the control group of plants without stressors. All analyses were replicated five times. All the data were analyzed by ANOVA and means were compared using a SNK test ($p < 0.05$) by Statistic version 14.0 (StatSoft, Tulsa, OK, USA).

## 3. Results

### 3.1. Gas Exchange Parameters

Since treatments differed significantly in their growth rate, the data is presented as a percentage with the control group of plants compared to the treatments for waterlogging, drought, and osmotic stress on an equivalent basis. The gas exchange parameters fluctuated within the treatment and between treatments, which disturbs the test of comparison of means between treatments (Figure 1). Thus, the most feasible method was to compare the gas exchange recorded on the first day or 0-day with its last ($k_{day}$) value recorded in each treatment and, when the average of $k_{day}$ was statistically different from the average 0-day, and comparisions were made to evaluate the measurerd inflection point of the descending curve of the evaluated parameters. With the control plants for comparison, $A_N$ decreased by 24.6%, in the plants measured at 12-days. However, $A_N$ recorded in the plants measured at 12-days ($12.4 \pm 1.0$ mmol $CO_2$ m$^{-2}$ s$^{-1}$) is not statistically different from 0-day ($16.4 \pm 2.2$ mmol $CO_2$ m$^{-2}$ s$^{-1}$; $p = 0.13$). The $A_N$ of waterlogged plants at-12-days was 41.2% lower than the $A_N$ of 0-day ($p < 0.0001$). In this way, photosynthesis was analyzed day-by-day to find the inflection point of the curve. So, in this treatment, 2 days have already shown a reduction of 30.1% ($p = 0.006$). The same was done for all treatments, where the plants at 8-days-drought-stress had a reduction of 98.6% and similarly to the previous 2 days without irrigation, was done showing a decrease in $A_N$ of 30.1% ($p = 0.034$). The degree of decrease in $A_N$ in the salt-stressed-plants was dependent of osmotic potential ($\Psi_s$). Thus, the plants measured at 12-days-of salt-stressed ($Y_s = -0.44$ MPa) showed a $A_N$ 51.4% lower than on 0-day; however, the inflection point of the curve occurred only after 8-days of salt-stress ($6.0 \pm 0.9$) compared to 0-day $A_N$ ($15.9 \pm 1.1$). Besides, the plants at 8-days-salt-stressed-($Y_s = -0.89$ MPa) showed a $A_N$ 90.4% lower than on 0-day plants, with the inflection point on the 2nd day ($p = 4.3 \times 10^{-4}$). Another form to point out the difference in the $A_N$ between treatments was demonstrated by adding up all $A_N$ recorded from 0-day to k-day $\left( \sum_{i=k}^{i=0} A_N \right)$. So, we have that the $\sum_{i=k}^{i=0} A_N$ was 520.22, 437.11, 233.00, 379.42, and 161.30, respectively in control, waterlogging, water stress, $Y_s = -0.44$ MPa, and $Y_s = -0.89$ MPa; values that generate a descending linear curve of the type y = $-98.4$ x + 643.9 (R$^2$ = 0.93). However, $\sum_{i=k}^{i=0} A_N$ of the treatments was control > waterlogging > $Y_s$ $-0.44$ MPa > drought-stress > $Y_s$ $-0.89$ MPa.

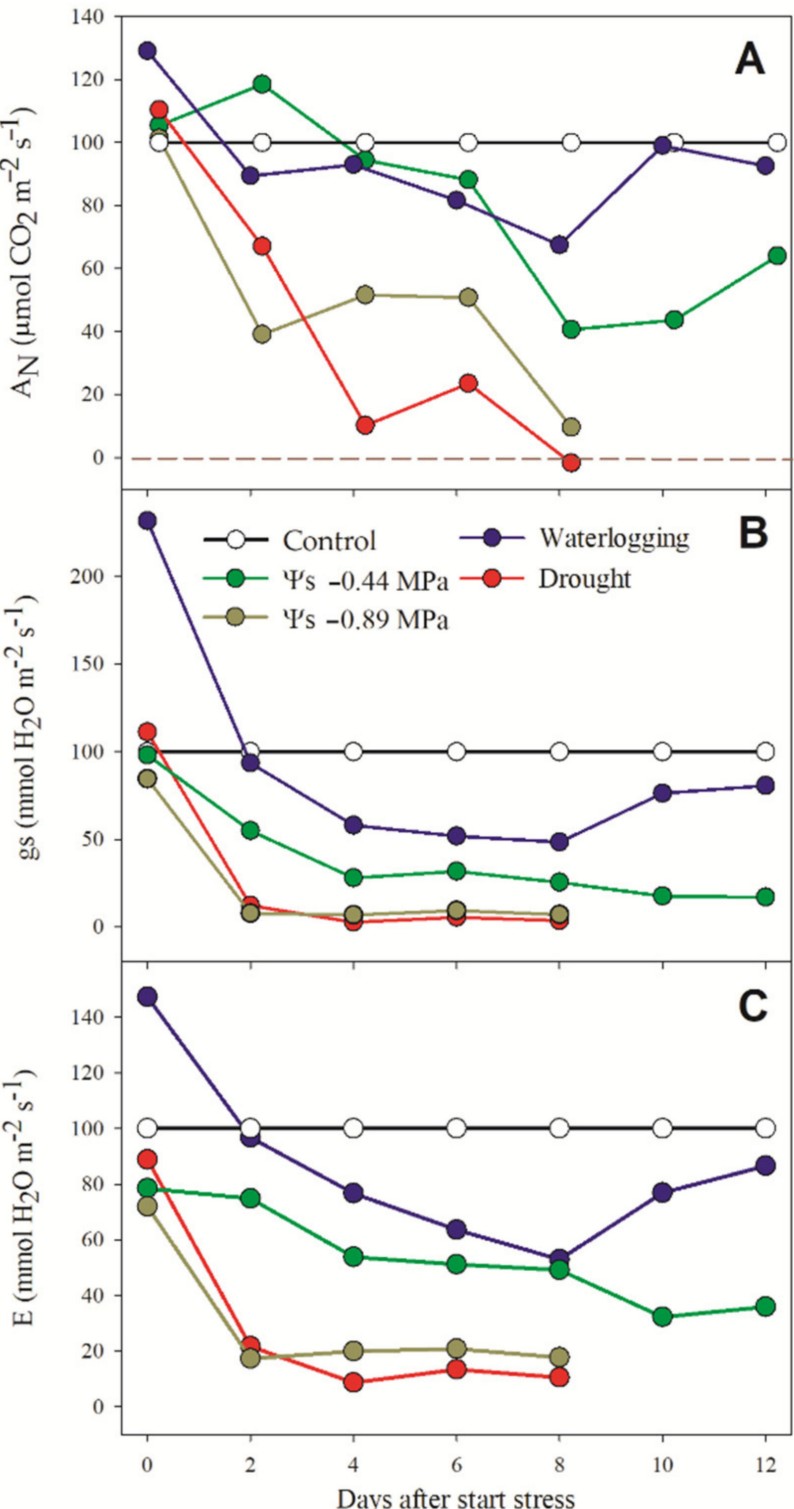

**Figure 1.** Net photosynthesis (**A**), conditions of stomatic conductance (**B**), and transpiration (**C**) measured in *Vigna unguiculata* under conditions of control, waterlogging, drought, salt stress $Y_s = -0.44$ MPa and $Y_s = -0.89$ MPa. The values are expressed as a percentage against the control group of plants, (horizontal line to 100%). The daily count was carried out from the beginning to the 12th day. The red and brown lines that are interrupted before the end of the experiment denote plants that did not support or failed the stressor conditions. Each point represents the average (±SE) $n = 5$.

Even though $g_s$ was strongly correlated with $A_N$ (detail in discussion) the $g_s$ profile was strongly distinct from the $A_N$ profile. In the control and waterlogging plants, the $g_s$ was 3.8-fold and 2.2-fold higher in 12-days-plants in comparison to 0-day-plants (Figure 1). Elsewhere, the plants at 8-days of-drought-stress and the plants at 8-days-of salt-stress ($Y_s$ = −0.89 MPa) showed a decrease in $g_s$ in the order of 86% and 73%, respectively. The plants measured at 12-days of salt stress ($Y_s$ = −0.44 MPa) showed a completely different profile, displaying a $g_s$ 2.5% lower than the control $g_s$ ($p$ = 0.88). So, the $\sum_{i=k}^{i=0} g_s$ was 29,540; 20,734; 8266; 2187; and 1768 with a linear descending curve with y = −7409x + 34,732 ($R^2$ = 0.92). However, $\sum_{i=k}^{i=0} g_s$ of the treatments following the same pattern was registered to $A_N$; control > waterlogging > $Y_s$ −0.44 MPa > $Y_s$ −0.89 MPa > drought-stress.

The transpiration (E) followed the same pattern of $g_s$ where plants measured at 12-days in thecontrol-group of plants and at 12-days-waterlogging showed E 55.1% and 31.6% higher than 0-day-plants (Figure 1C). The plants measured at 8-daysof drought-stress and the plants measured at 8-days ofsalt-stress ($Y_s$ = −0.89 MPa) showed a decrease in $g_s$ in the order of 74.1% and 57.4%, respectively. The plants measured at 12-days of salt-stress ($Y_s$ = −0.44 MPa) followed the same pattern registered to $g_s$, where 12-days-plants don't statistically differ from 0-day-plants ($p$ = 0.52). The $\sum_{i=k}^{i=0} E$ was the same as that registered to a $g_s$ with a linear descending curve with y = 61.92x + 321.74 ($R^2$ = 0.96).

### 3.2. Chlorophyll a Fluorescence Parameters

The fluorescence data are interconnected, counting a single pattern in different ways, these data will be presented by treatment, not by each analyzed feature. In all cases it's clear that waterlogged-plants did not have a physiological significance on all analyzed parameters (Figure 2). Although the optimal efficiency of PSII ($F_v/F_m$) measured in plants at 6-daysof waterlogging, 0.802 ± 0.001 was significant compared to control-plants (0.900 ± 0.001; Figure 2A). This difference, which in a subtle way became even more significant over time cannot be interpreted as a physiological difference for the plant. This understanding also translates to the quantum yield of photosystem II ($F_{PSII}$), which on 8-day was 20.5% higher than the control, but on 12-day was 33% lower (Figure 2B). An integrated analysis of the parameters shows that the electron transport rate (ETR; Figure 2C) registered in waterlogging-plants also fluctuated over the whole experiment; a fact that did not translate into increases or decreases in photochemical quenching ($q_P$; Figure 2D), nor in non-photochemical quenching ($q_{NP}$; Figure 2E), but in the extinction of all photons captured by the photosystems in the form of heat (D; Figure 2G) that was 40.6%, 26.0%, 19.6%, and 28.5% higher in waterlogging-plants when compared to control-plants.

In relation to the other treatments, made by integrating them in a systemic way, since the profiles presented by other abiotic-stressors show in salt-stressed-plants ($Y_s$ = −0.44 MPa). Thus, apart from drought-stressed-plants, the salt-stressed-plants ($Y_s$ = −0.44 MPa) show a decrease in $F_v/F_m$ in 14.2%, 19.7%, 22.6%, 27, 7%, 31.9% and 36.8% at 2, 4, 6, 8, 10, and 12 days after stress starts, respectively (Figure 2A). The profile was continuously decreasing with additional impact as the plants perceived the stressors and responded with a decrease in $F_v/F_m$. As stressors increased, the $F_v/F_m$ increased in the same proportion In a similar way, the $F_{PSII}$, ETR and qP, followed the same profile with a strong drop in the plants measured at 12-daysof salt-stress ($Y_s$ = −0.44 MPa) in order of 39.1%, 48.9% and 38.6%. These data are corroborated with more or less amplitude with $q_{NP}$, NPQ, D and PE. These last two factors were crucial in salt-stressed-plants ($Y_s$ = −0.89 MPa), where D and PE were on average 272% and 247% in relation to its control plants registered on the same day.

An interesting integration between the gas exchange data and the chloroplast electron transport chain (ETR) is presented in Figure 3, which form a regression curve, showing how stressed the plants are. We verified that there is a strong tendency of control-plants and waterlogged plants to show high stomatal aperture correlated with a lower ETR/$A_N$ ratio. At the same time, we verified a strong tendency of drought-stressed-plants and salt-stressed-plant ($Y_s$ = −0.89 MPa) to show a lower amount of stomata aperture and

higher ETR/$A_N$ ratio. Salt-stressed-plants ($Y_s$ = −0.44 MPa) also display an intermediate pattern of responses.

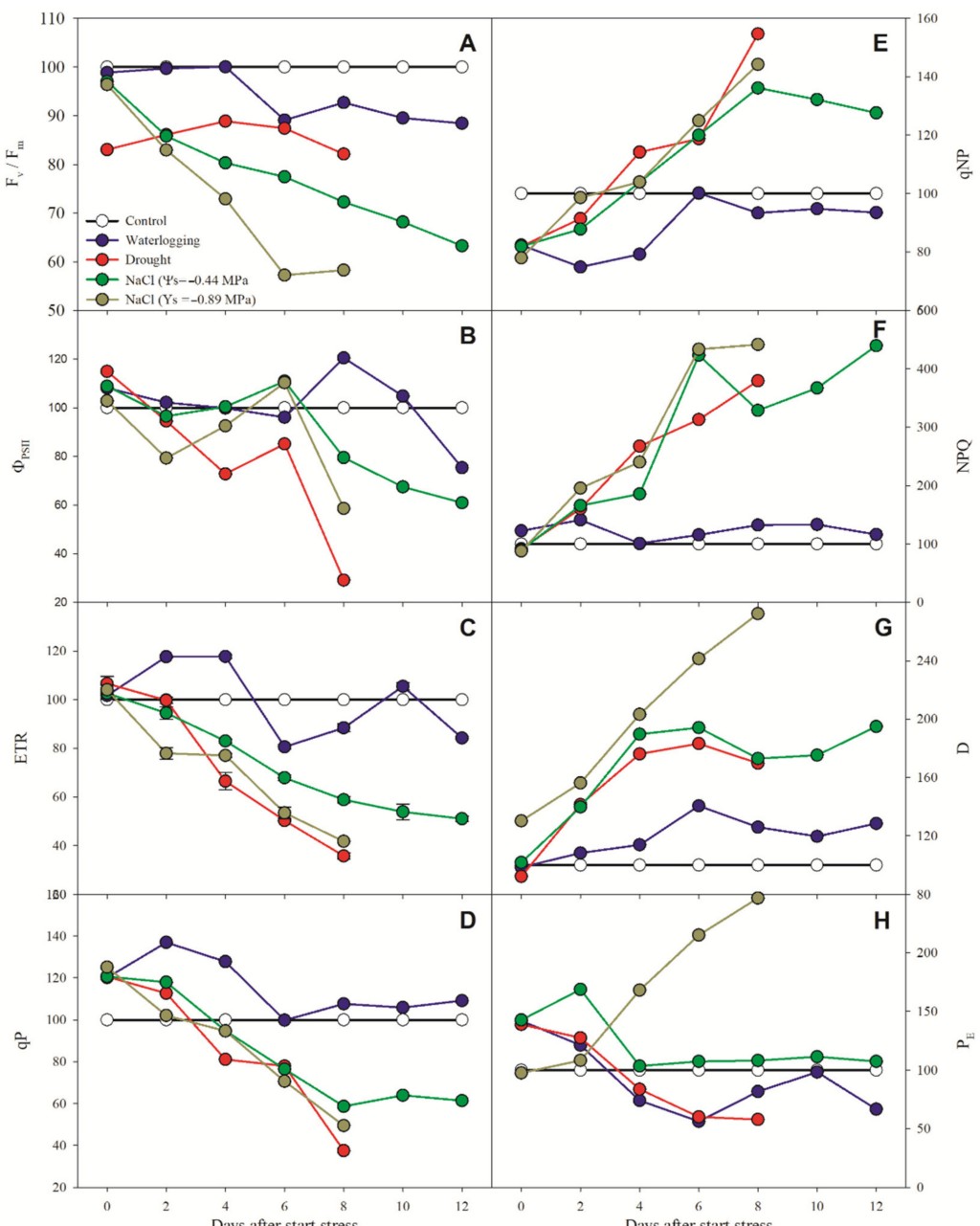

**Figure 2.** Optimal efficiency of PSII ($F_v/F_m$; (**A**)), quantum yield of photosystem II ($F_{PSII}$; (**B**)); electron transport rate (ETR; (**C**)); photochemical quenching ($q_P$; (**D**)); non-photochemical quenching ($q_{NP}$; (**E**)); total non-photochemical quenching (NPQ; (**F**)); fractions of absorbed photosynthetic active radiation dissipated as heat (**D,G**) and the fraction neither used in photochemistry nor dissipated thermally (PE; (**H**)) evaluated in *Vigna unguiculata*. The values are expressed in control (%), (horizontal line to 100%). The daily count was carried out from the beginning to 12th day. The red and brown lines that are interrupted before the end of the experiment denote plants that did not support the stress. Each point represents the average (±SE). *n* = 5.

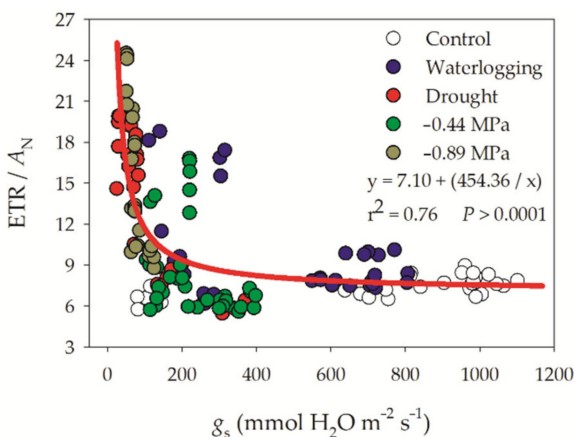

**Figure 3.** Relationship between the ratio of electron transport rate (ETR) and net photosynthesis ($A_N$) versus stomatal conductance ($g_s$) evaluated in *Vigna unguiculata* under control (white), waterlogged (blue), drought (red), $\Psi_s = -0.44$ MPa (green), and $\Psi_s = -0.89$ MPa (brown). Each data represent one plant evaluation in the whole experiment. Regression coefficient ($r^2$) and *p* are shown. *n* = 35.

### 3.3. Biochemical and Enzymatic Analysis

The determination of chlorophylls and carotenoids, which correlates to semi-direct information about the physiological conditions of plants under stress conditions. We show that in the control-plants, the contents of chlorophyll a, chlorophyll b and chlorophyll a + b are similar, i.e., more chlorophyll and less carotenoids, whereas, all plants experiencing stressors have less chlorophylls and more carotenoids (Figure 4). A comparison was made where the more stressed plant (drought) was compared with the control plants. The drought-stressed-plants showed a strong decrease in chlorophyll a (40.5%), chlorophyll b (60%) and chlorophyll a + b (46%); while at the same time the total carotenoids increased by 99.1% from $0.40 \pm 0.03$ g k$^{-1}$ DW to $0.80 \pm 0.04$ g k$^{-1}$ DW. Similarly, but distinctly, the salt-stressed-plants ($Y_s = -0.89$ MPa) showed a decrease in chlorophylls a (56.4%), chlorophyll b (73.5%) and chlorophyll a + b (68.6%), while increasing the total carotenoids by 12.5%, from $0.40 \pm 0.03$ g k$^{-1}$ DW to $0.45 \pm 0.13$ g k$^{-1}$ DW.

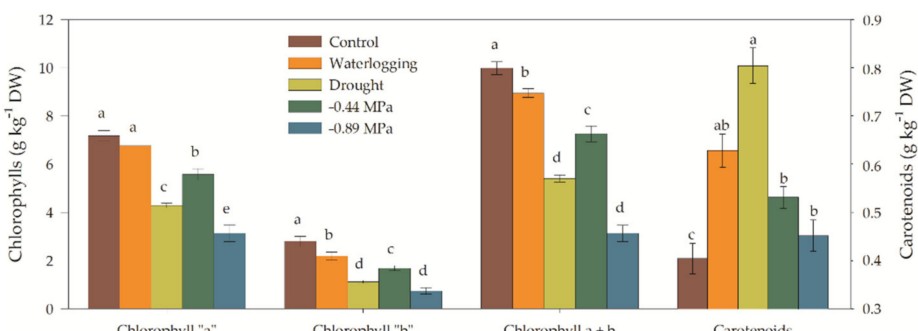

**Figure 4.** Chlorophyll "a", Chlorophyll "b", Chlorophyll "a + b", and total carotenoids evaluated in *Vigna unguiculata* under control, waterlogging, drought, $\Psi s = -0.44$ MPa, and $\Psi s = -0.89$ MPa. In each pigment, different letters denote statistical significance (*p* < 0.001). Each bar represents the average ($\pm$SE). *n* = 5.

To cope with a probable increase in the electron transport rate and its ineffective use as a reducing power to reduce the $NAD(P)^+$, the defense enzymatic system expressed a different intensity depending on the applied stressor (Figure 5). SOD activity was strongly increased by 392% in waterlogging-plants and by 128% in salt-stressed-plants ($\Psi$s = −0.44 MPa), moderately increased by 64% in drought-stressed plants and decreased 10%[ns] in salt-stressed-plants ($\Psi$s = −0.89 MPa) (Figure 5). In a similar manner, CAT activity was increased by 278% in waterlogging-plants, by 244% in salt-stressed-plants ($\Psi$s = −0.89 MPa), but less so in 126% in drought-stressed plants and in 85% in salt-stressed-plants ($\Psi$s = −0.44 MPa). The APX activity is also increased with the application of stressors. The increase in activity was strongly increased by 285% in salt-stressed-plants ($\Psi$s = −0.89 MPa) and 151% in drought-stressed-plants or 2.5-fold in waterlogging-stressed-plants and 79% in salt-stressed-plants ($\Psi$s = −0.44 MPa).

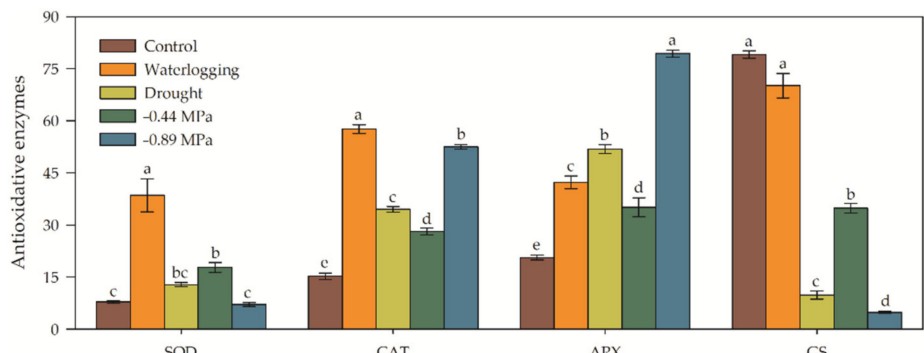

**Figure 5.** Superoxide dismutase (SOD), catalase (CAT), ascorbate peroxidase (APX), and glutamine synthetase evaluated in *Vigna unguiculata* under control, waterlogging, drought, $\Psi$s = −0.44 MPa, and $\Psi$s = −0.89 MPa. SOD, CAT, APX, and GS denotes, respectively U SOD $kg^{-1}$ protein, mmol $H_2O_2$ $min^{-1}$ $kg^{-1}$ protein, mmol ascorbate $min^{-1}$ $kg^{-1}$ protein, mol g-glutamyl-hydroxamate (gGH) $h^{-1}$ $kg^{-1}$ protein. In each enzyme, different letters denote statistical significance ($p < 0.001$). Each bar represents the average (±SE). $n = 5$.

GS activity, in turn, displayed a distinct pattern different from the other antioxidant system enzymes (Figure 5). While the salt-stressed-plants ($\Psi$s = −0.89 MPa) and drought-stressed-plants showed a sharp drop of 94% and 88%, the salt-stressed-plants ($\Psi$s = −0.44 MPa) showed a median drop of 56%. Waterlogging-stressed-plants showed a non-significant 11% drop in GS activity.

The stress factors associated with the different variables negatively affected the vitality of the plants, thus limiting the ability to convert sunlight into chemical energy (photosynthesis), the growth and dry weight of in the components of *V. unguiculata*. In summary, comparisons between stress factor reactions revealed different significant behaviors (Figure 6). Protein concentration in stressed-plants varied strongly among the plants with different stressors (Figure 6). While the salt-stressed-plants ($\Psi$s = −0.44 MPa) and the salt-stressed-plants ($\Psi$s = −0.89 MPa) showed a strong and median decrease of 77% and 15.5%, respectively in protein level; waterlogging-stressed-plants and drought-stressed-plants showed an increase of 26.4% and 7.5%[ns]. The concentration of $H_2O_2$, proline and malondialdehyde (MDA) are a direct result of stress level and indirectly results in differing amounts of activity of the enzymes of the antioxidative system (Figure 5). Based on this precept, it is verified that $[H_2O_2]$ was 2.3-, 1.6-, 1.5- and 1.9-fold higher than control, respectively in salt-stressed-plants ($\Psi$s = −0.89 MPa), drought-stressed-plants, waterlogging, and salt-stressed-plants ($\Psi$s = −0.44 MPa) (Figure 6). Peroxidation of membrane lipids, as measured by MDA accumulation, is also directly affected by severity of stress. We found that MDA increased by 98% in waterlogging-stressed-plants; while in salt-stressed-plants ($\Psi$s = −0.44 MPa), drought-stressed-plants, and salt-stressed-plants ($\Psi$s = −0.89 MPa) the increase was 405%, 650%, and 740%, respectively. Of the biochemi-

cal/enzymatic compounds, proline production and accumulation was the one biochemical compound that fluctuated the least between treatments, from a slight increase of 24.1%[ns] to a significant increase of 72.7%, respectively in waterlogging-stressed-plants and drought-stressed-plants. Unlike, in salt-stressed-plants ($\Psi s = -0.44$ MPa) and in salt-stressed-plants ($\Psi s = -0.89$ MPa) which showed a mild (8.4%[ns]) and moderate (52%) decrease in proline production and accumulation (Figure 6).

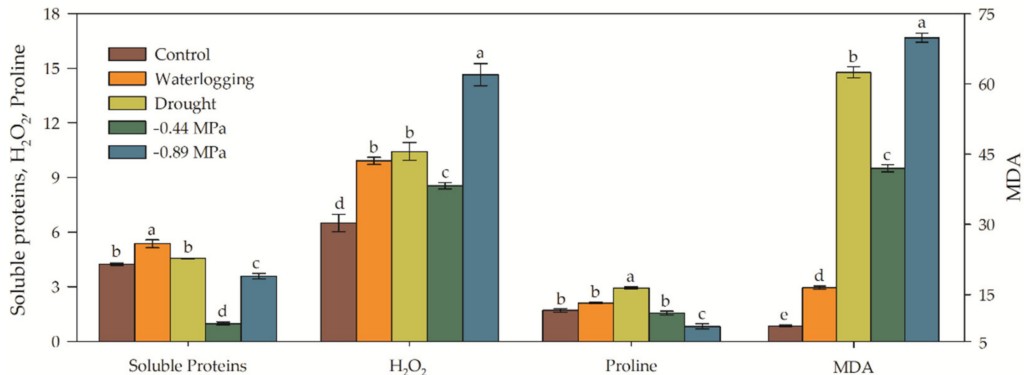

**Figure 6.** Soluble proteins (g protein kg$^{-1}$ DW), hydrogen peroxide (H$_2$O$_2$; mmol H$_2$O$_2$ kg$^{-1}$ DW), proline (mmol proline kg$^{-1}$ DW), and malondialdehyde (MDA; mmol MDA kg$^{-1}$ DW) evaluated in *Vigna unguiculata* under control, waterlogging, drought, $\Psi s = -0.44$ MPa, and $\Psi s = -0.89$ MPa. In each metabolite, different letters denote statistical significance ($p < 0.001$). Each bar represents the average ($\pm$SE). $n = 5$.

### 3.4. Total Biomass and Leaf Area Estimation

The imposition of progressive stressors for 12 days reduced biomass to a greater or lesser extent, all stressors caused a decrease in plant biomass as well as in total leaf area (Figure 7). The measure of the dry root mass was 66.1%, 70.0%, 71.1%, and 75.1% lower, respectively, in salt-stressed-plants ($\Psi s = -0.44$ MPa), drought-stressed-plants, salt-stressed-plants ($\Psi s = -0.89$ MPa), and waterlogging-stressed-plants. The root profile did not translate in the same order for shoot biomass and leaf biomass. Shoot biomass was 28.8%, 33.8%, 49.6%, and 54.8% lesser, respectively in salt-stressed-plants ($\Psi s = -0.44$ MPa), drought-stressed-plants, waterlogging-stressed-plants, and salt-stressed-plants ($\Psi s = -0.89$ MPa). Dry leaf biomass and total leaf area were the two biometric characteristics that most displayed the impact of the stressors. In this sense, we registered a fall in leaf in biomass 52.3% (drought-stressed-plants), in 56.5% salt-stressed-plants ($\Psi s = -0.89$ MPa), in 62.5% (waterlogging-stressed-plants), and 96% salt-stressed-plants ($\Psi s = -0.44$ MPa). A distinct for the different stresors was observed in the total leaf area; where salt-stressed-plants ($\Psi s = -0.89$ MPa), salt-stressed-plants ($\Psi s = -0.44$ MPa), drought-stressed-plants, and waterlog-stressed-plants showed a significant reduction in leaf biomass in the order of 53%, 56%, 57%, 73%. A distinct profile was recorded for plant height, where waterlog-stressed-plants, salt-stressed-plants ($\Psi s = -0.89$ MPa), salt-stressed-plants ($\Psi s = -0.44$ MPa), and drought-stressed-plants presented a shoot height of 10.6%, 24.0, 30.0, and 43.0 lower than that recorded in non-stressed plants (Figure 7).

### 3.5. RT-PCR

All amplifications resulted in amplicons of the expected length, of the thirteen genes in the four abiotic stress conditions (waterlogging, drought and two osmotic potentials compared to the control). The measured sets of values for the various genes in the plants under the stressors of drought and the two osmotic potentials due to NaCl were generally larger values. The differences between the treatments affected by osmotic potentials were detected with an increase in the expression of some genes (Figure 8). Mainly the *VuCPRD14*, *VuCPRD65*, *VuDREB2* and *VuHsp17.7* genes as well revealed in Figure 8. However, in waterlogging conditions, we did not experience an increase in the expression of oxidative

stress genes, being very similar to the control. The expression genes results do not show significant differences between the abiotic stress treatments due to drought and the two with osmotic potentials provided by NaCl, which displays the role of the antioxidant enzymes, as well as shown in Figure 5. More details, see Figure 8.

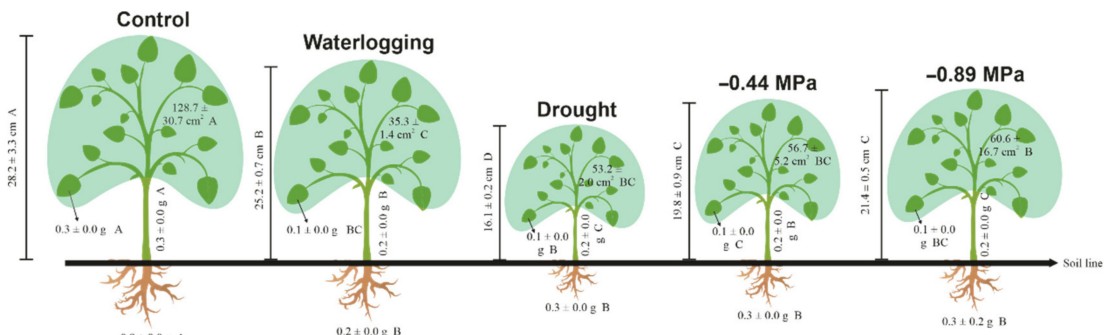

**Figure 7.** Root, shoot, leaves biomass, total leaf area, and plant height measured in *Vigna unguiculata* under control, waterlogging, drought, Ψs = −0.44 MPa, and Ψs = −0.89 MPa. In each analyzed features, different letters denote statistical significance ($p < 0.001$). Each bars represents the average (±SE). $n = 5$.

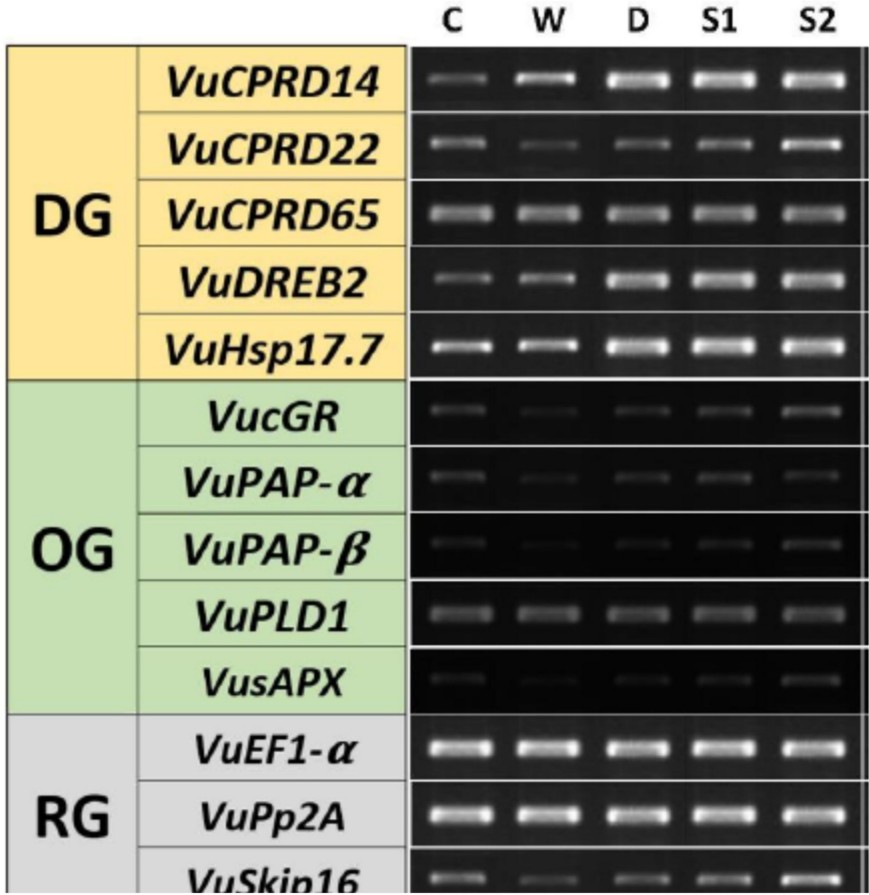

**Figure 8.** Comparative results of gene expressions of the evaluated treatments, based on RT-qPCR of five genes related to the imposition of drought (DG), five genes related to oxidative stress (OG) and three reference genes (RG). Control (C), waterlogging (W), drought (D), Ψs = −0.44 MPa (S1), Ψs = −0.89 MPa (S2).

## 4. Discussion

Water stress is the most important constraint for crop production. It may be determined by measuring physiological parameters such as net photosynthesis, stomatal conductance, transpiration, and leaf temperature among others. Also available are the biochemistry found by measuring the chlorophyll and carotenoids content, soluble protein, antioxidant enzymes, $H_2O_2$, MDA and proline production and accumulation [33]. Additional characteristic factors can be used to detect stress, such plant height, dry biomass, and leaf area, or empirically as plant architecture–plant health (leaf yellowish, chlorosis, necrosis, and leaf abscission). Plants that have been placed under excessive stressor situations may show a slow rate of recovery of the PSII antenna system from the excited to the unexcited state, which implies a lower rate of $CO_2$ fixation, which is transiently reduced by the NPQ [6].

Net photosynthesis, stomatic conductance, and transpiration rates dropped sharply in stressed-cowpea plants compared to non-stressed plants. Similar patterns were previously reported in this species [36]. Independently of the other studied stressors, both water and osmotic stress, analyzed in this study, compared with their control counterparts, exhibited a decrease in all gas exchange parameters. This fact has a strong positive correlation between $A_N$ and $F_v/F_m$ ($r = 0.954$; $p = 1.60^{-21}$), $F_{PSII}$ ($r = 0.989$; $p = 8.11^{-21}$), ETR ($r = 0.966$; $p = 5.20^{-15}$), Chla ($r = 0.902$; $p = 1.77^{-11}$), Chlb ($r = 0.868$; $p = 1.83^{-8}$), and negatively correlated with MDA ($r = -0.959$; $p = 5.1^{-13}$) and APX ($r = -0.798$; $p = 1.81^{-6}$). The physiological measure of $g_s$ should be used to distinguish between cultivars which have a drought avoidance mechanism or a drought resistance mechanism [37,38]. In fact, $g_s$ also was positively correlated with all growth parameters like RBB ($r = 0.521$; $p = 7.62^{-3}$), SDB ($r = 0.867$; $p = 2.04^{-8}$), LDB ($r = 0.388$; $p = 0.052$), and PH ($r = 0.761$; $p = 1.02^{-5}$), except leaf area ($p = 0.07$; Figure 9). Therefore, the physiological measure of $g_s$ should be used to discriminate cultivars having drought avoidance or drought resistance mechanisms [37,38]. Plants subjected to waterlogging for a long time can display a decrease in photoassimilates, which can also significantly reduce metabolic activity. This leads us to argue that the decrease in root biomass, evidenced by the measured 75.1% reduction, could be a consequence of a lower metabolic rate in the roots under hypoxia, a reduction of mitochondrial aerobic respiration, and subsequent oxygen deprivation, as reported by Awala, et al. [39]. NaCl present in the soil reduces the osmotic potential (it becomes more negative) and prevents the absorption of water and mineral elements, and consequently, the weakening of the roots, as found in this study, where there was a reduction in the dry mass of the roots and other organs of the plant [40,41]. Salinity, in turn, can provoke inactivation of some enzymes in the Calvin-Benson cycle, causing a downregulation of photosystem and then ROS production [40,41].

A possible chlorosis in the leaves of the waterlog-stressed-plants was not visually registered but quantified directly by a decrease in the levels of chlorophylls a, and b. Also, visually, the waterlog-stressed-plants show the most substantial reduction in their leaf area, from $128.7 \pm 30.7$ in the control plants to $35.3 \pm 1.4$ (a reduction of 73% in the waterlog-stressed plants. El-Taher et al. [20] described that decreased photosynthetic pigments in cowpea plants under high levels of salinity are due to increments of the chlorophyllase enzyme [20,42]. In fact, waterlogging resulted in visible yellowing and premature senescence of leaves, and greater decline in chlorophyll content, and membrane stability. These issues may be proved from an inverse correlation between Chla and NPQ ($r = -0.488$; $p = 0.0134$), $H_2O_2$ ($r = -0.639$; $p = 5.78^{-4}$), MDA ($r = -0.945$; $p = 1.26^{-12}$), and APX ($r = -0.85$; $p = 3.86^{-8}$) (Figure 9).

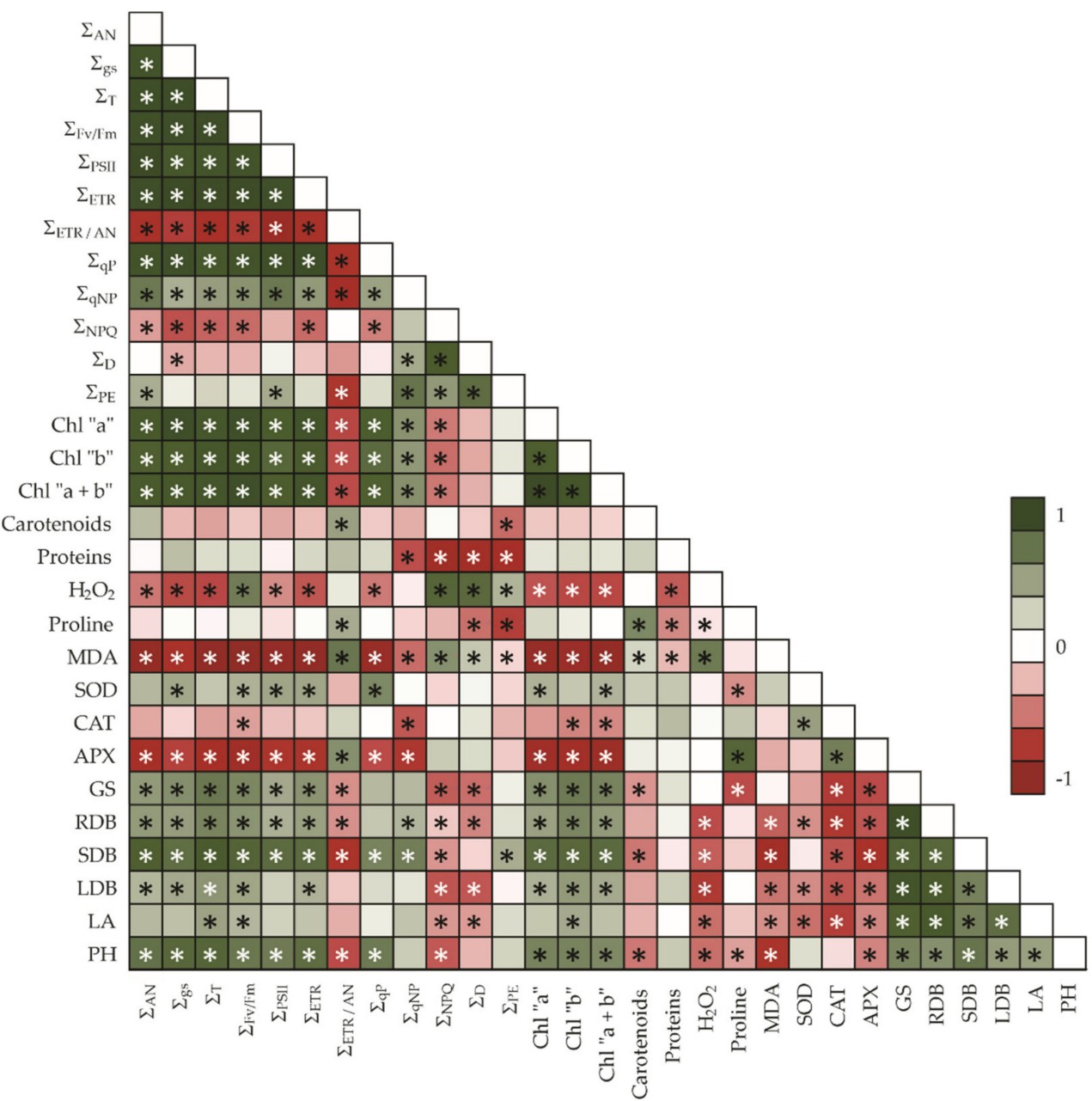

**Figure 9.** Pearson's correlation among all analyzed features where a cooler color denotes a positive correlation, while warmer colors denote a negative. The Pearson's analysis was done with 5 repetition. The false colors were obtained as legend. The asterisks denote the significant correlation ($p \leq 0.05$). $S_{AN}$. Sum of all net photosynthesis values; $S_{gs}$. Sum of all stomatal conductance values. $S_T$. Sum of all transpiration values; $S_{Fv/Fm}$. Sum of optimal efficiency of PSII; $S_{FPSII}$. Sum of all quantum yield of photosystem II values; $S_{ETR}$. Sum of electron transport rate values; $S_{qP}$. Sum of photochemical quenching values; $S_{NPQ}$. Sum of non-photochemical quenching values; $S_{qNP}$. Sum of non-photochemical quenching values; $S_{NPQ}$. Sum of total non-photochemical quenching values; $S_D$. Sum of fractions of absorbed photosynthetic active radiation dissipated as heat values; $S_{PE}$. Sum of fraction neither used in photochemistry nor dissipated thermally values; Chl "a". Chlorophyll "a"; Chl "b". Chlorophyll "b"; $H_2O_2$. Oxygen peroxide; MDA. Malondialdehyde; SOD. Superoxide dismutase; CAT. Catalase; APX. Ascorbate peroxidase; GS. Glutamine synthetase; RDB. Root dry biomass; SDB. Shoot dry biomass; LDB. Leaf dry biomass; LA. Leaf area; PH. Plant height.

Chlorophyll a fluorescence parameters are often used to study the damage induced by different types of stress in photosystem II (PSII), which is useful, quantitative, rapid and non-invasive to understand the different mechanisms and related aspects of photosynthesis. In addition, fluorescence measurement is considered a direct and powerful measure of the functional state of photosynthesis. In the different levels of stress, the photochemical variables, a decrease in the rate of electron transport (ETR) in the treatment of maximum salinity ($\Psi$s = $-0.89$ MPa) is presented, also a ratio of ETR/$A_N$ is large especially in the treatment of drought (Figure 2), this biochemical mismatch causes the inhibition of the electron transport chain, leading to an excessive accumulation of ROS.

To cope with stress the plants have a complex system to protect against oxidative damage. Chlorophyll photobleaching which leads to fewer photons directed to the photosystem I, and less reductant power is produced, leading to less ROS production. The results presented in this study, shows that under abiotic stress, *V. unguiculata* could promote chlorophyll photobleaching as previously reported in this species [36,43]. Chlorophyll photobleaching is also reported in *Brassica* sp. [44], *Coffea arabica* [32], *Salix* sp. [45], *Capsicum annuum* [46], and *Vigna radiata* [47]; all species which suffer under abiotic stress.

Enhancement of qE may offer increased photoprotection under high ETR because the enhancement promotes heat dissipation [6,32,48]. Violaxanthin deepoxidase (VDE) is an enzyme that uses ascorbate in the reduced form as a cofactor for the production of zeaxanthin [49]. Lumenal VDE is activated by the reduction in pH as a consequence of ETR. Thus, with a photosynthetic rate reduced by stress, less ATP will be reoxidized to ADP, causing other upstream molecules to remain reduced. It is in this way that VDE can act, as it uses reduced ascorbate to release excess reducing power in the form of heat, thus avoiding PSI photoinhibition [6].

However, plants have extensive defense machinery; both enzymatic or non-enzymatic like carotenoids [50] and proline. As a result of this, plants activate enzymatic and non-enzymatic defense system. In the present study, both enzymatic (SOD, CAT, APX, and GS) and nonenzymatic components (total carotenoids, $H_2O_2$, MDA, and proline) increase with all plant stressors. The increase of enzymatic and non-enzymatic protector systems consequently protecting plant tissues against oxidative injury are evidenced by decreased amounts of MDA, $H_2O_2$, as previously reported in V. unguiculata [36]. Superoxide radical and hydrogen peroxide content increased in salt-stressed-cowpea plants [36]. The increase of CAT in response to water stress aligns with previous studies with cowpea, whereas Nair, et al. [51] described a decrease in peroxidases in water-sensitive-cowpea plants. SOD, CAT, and APX activities was increased in cowpea plants under 3.5 or 7 dS m$^{-1}$ [36]. The second line of antioxidant defense is activated in more highly stressed plants with the regulation of ascorbate peroxidase [52]. This enzyme would oxidize the $H_2O_2$, using reduced ascorbate as a cofactor, in turn generating monodehydroascorbate, and afterwards dehydroascorbate. The dehydroascorbate is then reduced to ascorbate by action of dehydroascorbate reductase with an expenditure of 1 mol of reduced glutathione or 1 mol of NADPH [53,54]. The formation of reduced ascorbate involves the expenditure of 1 mol of dehydroascorbate, 1 mol of reduced glutathione, and 1 mol of NADPH to finally be reduced to ascorbate. At sufficiently high concentration, ascorbate could complement SOD in superoxide removal [55,56]. Another ascorbic acid pathway involves a glycolysis metabolic deviation of UDP-glucose that after many reactions could produce D-glucoronate through the action of glucuronate reductase using NADPH as cofactor [30].

Another line of defense is through the synthesis and accumulation of proline. Proline accumulation is an important criterion for selecting drought tolerant crops [57,58], including cowpea [59,60]. While $H_2O_2$ and MDA production and accumulation might be an important criterion to selecting the drought sensitive plants [51,61,62]. In accordance with Berteli et al. [57] the proline synthesis would be promoted by the GS/GOGAT cycle (highlighted in orange in Figure 10). In accordance with these authors, glutamate concentration was 137% and 176% greater, respectively in 60 mM and 120 mM NaCl. Under conditions of osmotic stress, the GS/GOGAT cycle has been shown to be the most important source

of glutamate. Although GS was not affected by the salt stress, Fd-GOGAT activity was 2-fold higher in salt-stressed-cowpea plants, as compared to the control plants [57]. The induction of Fd-GOGAT synthase under salt stress may provide the glutamate required for the proline synthesis which is a common response to salt stress [57]. In contrast, an increase in glutamate dehydrogenase (GDH) NAD(P)H dependence was observed [63] and consequently more glutamate was produced, which may be used to biosynthesize proline. GDH activity increased with increasing concentration of NaCl up to 200 mM NaCl and 800 mM NaCl caused complete loss of GDH activity from sensitive cultivar but not from tolerant cultivar [64]. Studies indicate that the increase in GDH is promoted by a toxic effect of increased endogenous level of ammonia which probably accumulates due to efficient $NO_3^-$ reduction [63,65], while GS-GOGAT cycle is the normal source of glutamate in plants [57]. Proline accumulation seems to be the main mechanism responsible for its drought tolerance in cowpea plants [57,63,66,67]. In cowpea plants measured at 9-days-of waterlog stress showed an increase in 97.3% in proline when compared to control [67].

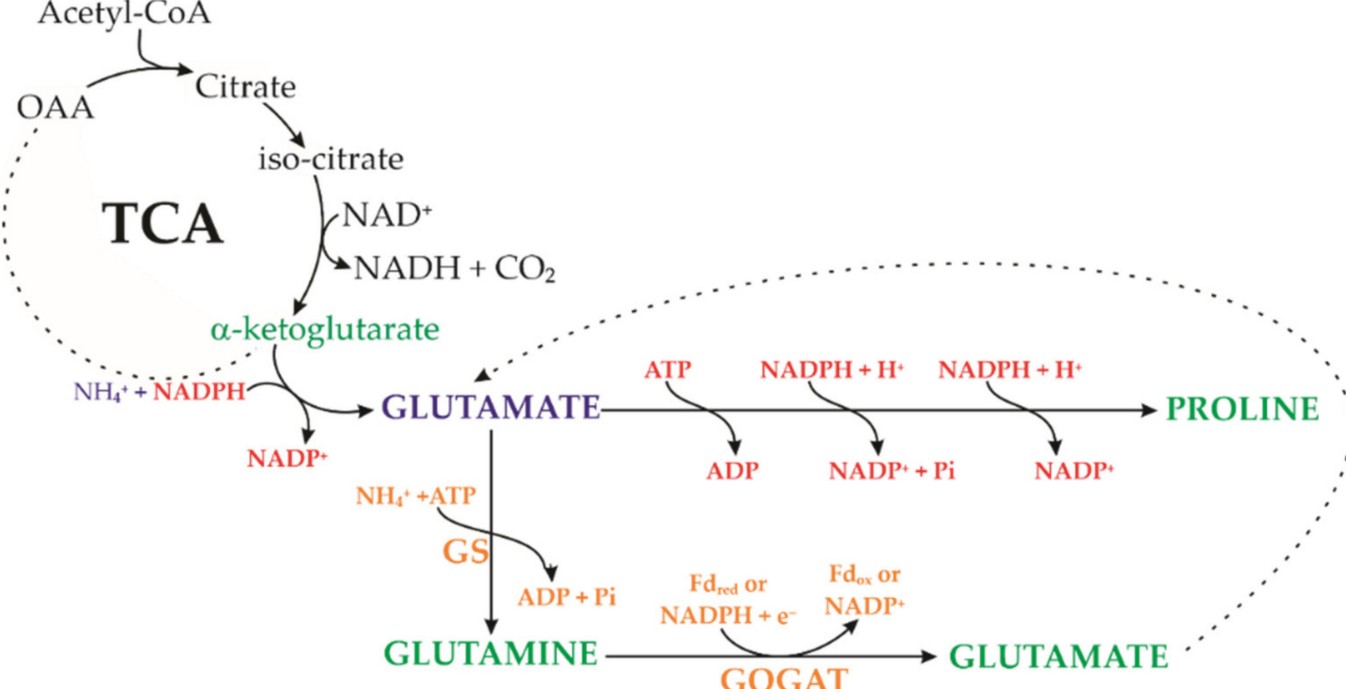

**Figure 10.** Proline biosynthesis comprises a complex cascade of oxidation-reduction reactions, with a-ketoglutarate (KG) shifting from the tricarboxylic acid cycle (TCA). KG suffers two biochemical steps spending NADPH to produce glutamate (Glu). Glutamate undergoes 4 steps to be converted to proline, namely: (1) addition of phosphate by glutamate 5-kinase, with the consumption of 1 mol of ATP; (2) reduction performed with the participation of two moles of NADPH, by enzyme glutamate dehydrogenase and pyrroline-5-carboxylate reductase which uses one mole of $FADH_2$ as a cofactor. So, the final steps in proline (Pro) synthesis involve the reaction of 1 mol of glutamate, 1 mol of ATP, 2 mol of NADPH, and 1 mol of $FADH_2$ to form 1 mol of proline. An alternative metabolic pathway involves production of Glutamine (Gln) from Glu at expense of ATP. Afterwards, Gln would be reduced for produce newly Glu, via GS/GOGAT cycle. Glutamate may be reentered in a second Pro biosynthesis. Schematic figures adapted from literature data [68–70].

Many scholars defend that the chlorophyll a fluorescence as the first stressor signal [47,60,71,72] while the production of $H_2O_2$ and MDA are the stress signals, informing that the antioxidant systems are not able to cope with the ROS and the damage to the membrane phospholipids [66,73,74]. $H_2O_2$ has the capacity to initiate lipid peroxidation and degrade proteins. Figure 6 shows a slight decrease in protein levels as $H_2O_2$ increases;

a fact confirmed by moderate negative correlation (r = −0.605; *p* = 0.001) between these two features (Figure 9). Similar results were previously reported also by Nair et al. [51].

Previous works reported that the oxidative stress genes evaluated are elevated in susceptible cowpea genotypes compared to tolerant ones [27,75]. A putative function of *VuCPRD14* is associated with anthocyanin and medicarpin synthesis which protect plant cells from stress conditions, having a role in the dehydration response [27,76]. Da Silva et al. [75] describe that *VuCPRD14* gene may also be related to antioxidant metabolism against ROS formation. An increase of more than twice the value of some DG genes was associated with waterlogging stress treatments [27]. The *VuCPRD65* was described as an 9-Cis-epoxycarotenoid dioxygenase 1 associated to water stress [27] linked or not to the progressive dehydration associated with ABA synthesis [75] or involved with neoxanthine cleavage system [76], and probably associated with heat dissipation in the VAZ cycle [76,77]. Both *VuDREB2* and *VuHSP17* are calcium-dependent kinase proteins, in response to thermic stress and water stress [75,78]. Also, this gene was reported as initiator of dehydration expression responses under individual heat, drought, and combined drought and heat stress in root of two banana genotypes [79] or linked to salt tolerance *Triticum aestivum* [80]. Zea mays under severe and intermediate water deficit stress, compared to normal irrigation, showed *DREB-2* with higher values with 690 and 211% respectively [81]. In accordance with these authors the *DREB-2* may also be related to catalase activity, by analogy in response to ROS. *VuHSP17.7*, an sHSP family class I protein, was highly induced by high-temperature stress in *V. unguiculata* nodules, suggesting a role in signaling pathways under heat stress [78], while *GmHSP17.1*, encoding an sHSP in cytoplasm, was discovered and its values revealed by qRT-PCR and promoter-GUS analysis in nodules indicated that *GmHSP17.1* was specifically expressed in nodules [82], unlike our study that found this gene expressed in leaves, which follow the same line described to *Primula forrestii* [83] that describe that this gene is ubiquitously expressed in different plant organs. In accordance with these authors *PfHSP17.1* is a member of the plant cytosolic class which is highly up-regulated in the leaves of *P. forrestii*. In Cp5051 cowpea genotype the *VuHsp17.7* and *VuCPRD14* were described to be drought tolerant [27]. The evaluated values of these two genes were not statistically different from the control, it indicates that the variety Caupicor 50, used in this study may not be water stress tolerance because this treatment was the first to show a visual stress phenotype.

The results highlight the importance of studying both the biotic and abiotic events that guide the cultivation of a species [84]. Cowpea has been extensively studied, as there are many articles published about this species, testing the most diverse stressors and promoters of photosynthesis. Each article has it's own objectives and goals. However, we do not find in the literature an article that has discussed three types of stressors with in-depth, robust analyses, and with discussions that really consider the cellular metabolism as a whole, unfragmented system. The differentiation of this study is precisely that it integrates in a single study the three main abiotic stresses and discusses them with the depth that the topic demands. Figure 11 summarizes all analyses presented in this study in an integrated and multifactorial analysis. However, there is still a lot to be studied to fully elucidate the photosynthetic process, the dissipation of fluorescence and heat as a deepening of the analysis of carotenoids with the participation of chlorophyll a fluorescence by imaging [85,86] and the study of carotenoids violaxanthin, antheraxanthin, neoxanthine and zeaxanthin [32,87–90] to fully elucidate how the cowpea manages to dissipate excess energy in the form of fluorescence and heat.

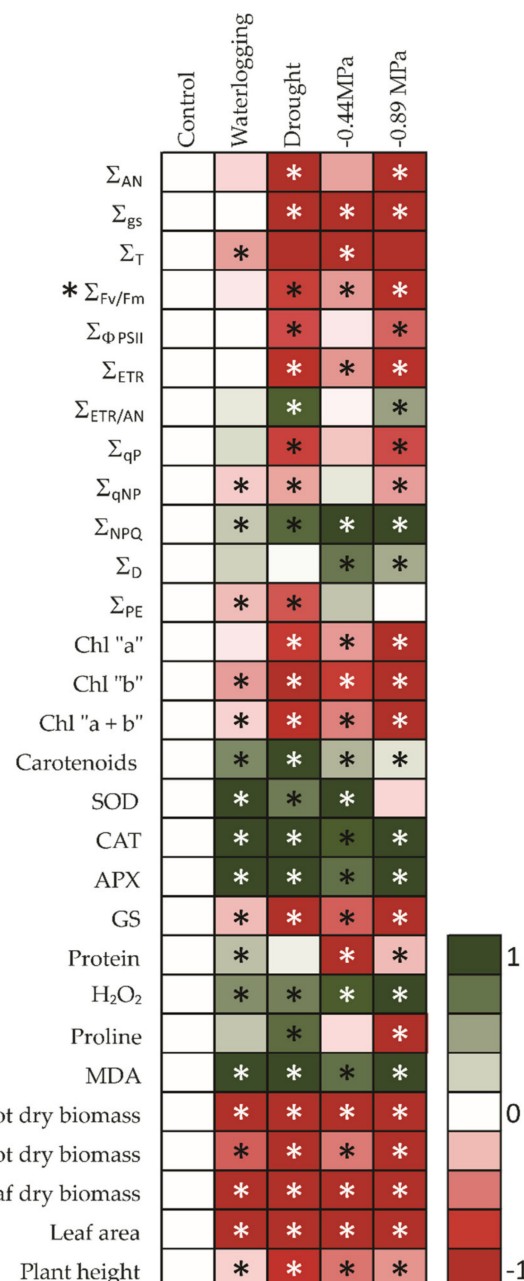

**Figure 11.** The heatmap shows an overview of all analysis features compared to each other. In each line the asterisks (*) denote significant results in relation to its control group of plants. The false colors were obtained by ratio between treatments and its controls (1st column) and then after $\log_2$ as a legend in right panel, where warmer colors denote a decrease in relation to control, while cooler colors denote an increase in each characteristics when compared to its controls. Each value was done by media (*n* = 5).

## 5. Conclusions

The rapid and continuous decrease in gas exchange, in line with the inverse situation in the emission of fluorescence and activation of downstream metabolic processes, the expenditure of a power reductor, a reduction of photosynthetic efficiency, a reduction of the concentration of chloroplast pigments, a strong activation of the defense system, proline synthesis and the exacerbated production of molecules elicited by excess reducing power, such as $H_2O_2$ and MDA, lead us to argue that cowpea has a robust defense system that is often able to block the harmful effects of abiotic stresses. Thus, we present here

an integrative analysis of the main physiological, biochemical, and molecular processes activated by *V. unguiculata* to protect itself from the exacerbated production of reducing molecules and prevent further damage from being triggered irreversibly. However, it is a fact that some types of stress, especially those added more abruptly, cause strong damage to biological structures without which the plant cannot recover, which then triggers events that culminate in programmed cell death.

**Author Contributions:** All authors contributed equally. All authors have read and agreed to the published version of the manuscript.

**Funding:** This research doesn't receive any financial input.

**Institutional Review Board Statement:** Not applicable.

**Informed Consent Statement:** Not applicable.

**Data Availability Statement:** Not applicable.

**Acknowledgments:** The authors thanks to the Facultad de Agronomía, Universidad de Córdoba, Montería, Colombia for permit its experiments and the anonymous reviewers that improved this paper.

**Conflicts of Interest:** The authors declare no conflict of interest.

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
