# Peer review of "Can Chlorophyll a Fluorescence and Photobleaching Be a Stress Signal under Abiotic Stress in Vigna unguiculata L.?"

_sustainability, doi:10.3390/su142315503_

Round 1

Reviewer 1 Report

Please refer to attachment for comments.

Author Response

RESPONSE TO REVIEWER 1

Reviewer 1: The authors provide a comprehensive study on the response of cowpea to drought, waterlogging and saline stress. In general, the manuscript is too long and can significantly shortened to concentrate on the salient findings from the study. This is particularly evident in the results and discussion which lack organisation and a logical progression of ideas. Too much emphasis is placed on a review of the findings from published studies rather than explaining the results obtained in the present study and relating this to published reports. Overall, the English requires revision as sentence construction needs to be improved it is often difficult to understand what is presented and it is grammatically incorrect. It is acknowledged that the authors have contributed a substantial amount of effort in this study, but the manuscript in its present form lacks focus. The work can be published if improvements are made.

Authors: The authors are grateful for your valuable contributions to improve this manuscript. We confirm that some efforts were done do be reduce the introduction and discussion. About English, we confirm that the text was corrected by a Native English Speaker (Jeffrey A. Brandley from Blue Springs, MO, USA) at 10 Oct 2022.

Reviewer 1: L1-3: the title is grammatically incorrect and needs to be refined

Authors: The authors are grateful for your valuable contributions to improve this manuscript. The

title has been reconstructed (Can Chlorophyll a Fluorescence and Photobleaching be a Stress

Signal Under Abiotic Stress in Vigna unguiculata L.?) and we hope it will please the reviewer

and make it clear to Sustainability readers

Reviewer 1: Introduction: suggest shortening the text

Authors: Thanks so much for your analysis and suggestions. Some sentences were excluding the introduction. Before the introduction has 1345 words, now 773, i.e., a shortness of 42.53%

Reviewer 1: L39: the first statement is very extreme, suggest revision

Authors: We've made an adjustment to the sentences; we hope it's clear to the reviewer. Please see it on line 38.

Reviewer 1: L50-58: The logic here does not make sense

Authors: All these sentences were deleted. Thanks for your appointment.

Reviewer 1: L83: drought cannot be due to excess

Authors: The authors apologize for this lapse. The correction can be seen in line 64.

Reviewer 1: L109-113: contradiction in these two sentences the first states that cowpea is tolerant to drought and the next that it is moderately salt sensitive

Authors: Dear reviewer, in these sentences, two previous studies are cited that affirm what was described in these studies. However, we searched for more articles and found that cowpea is drought tolerant, however there are genotypes with a greater or lesser degree of drought tolerance. Regarding salinity tolerance, some authors describe V. unguiculata as sensitive to salinity and others as tolerant. However, the degree of tolerance is usually cultivar or variety dependent. So, this information was added in the manuscript, on lines 78-79.

Reviewer 1: L189: this heading needs to be modified to be grammatically correct

Authors: The authors apologize for this lapse. The correction can be seen in line 148.

Reviewer 1: The gene expression results are missing

Authors: The authors apologize for this lapse. The correction can be seen in line 417-420, page 12.

Reviewer 1: Discussion: too long, shorten to focus on contextualising the results from the current study in relation to published literature, what is currently presented is more like a review of the literature. The results of the correlation analyses can be better explained so that the relationship between the data sets, and what the data actually means, can be conveyed to the reader. A more concise argument can be presented.

Authors: Dear reviewer, a significant portion of the discussion (50.54%) has been removed, making it cleaner and easier to read. Originally, the discussion had 3,306 words and in this new version, the discussion has 2,196 words. This shortness version is four pages less than original text. However, it should be noted that much of the discussion is used to present the antioxidant system, in special to detoxify reducing molecules and the role of proline, which insistently many authors still attribute to it the role of osmoregulator, forgetting that this role has already was revised and currently proline appears much more as a result of stress than as a way to avoid it.

Reviewer 1: The graphical depiction at the end of the manuscript requires revision, surely the salt stress is imposed through the soil (given climate change scenarios), H2O2 is not present in the soil as a stress signal and only leaf samples were analysed so the role of the roots in signaling, etc. was not evaluated.

Authors: The authors point out that the way it could give that impression; thus, in place of the soil, only a red line was inserted, which indicates that there was a cellular signaling, from the finest and absorbing roots to the central xylem, which then distributes this signal of stress to the leaves. In the present scheme a leaf is detached from the plant to show the cellular production of H2O2 and the defense mechanisms. We hope the changes please the reviewer. The authors apologize for this lapse.

To summarize, we have attempted to address all comments and have corrected all major and minor mistakes
outlined by the reviewer in the previous version our manuscript.

We truly believe that our manuscript is of suitable quality for publication in Sustainability.

Thank you for your attention.

Sincerely yours,

Dr. Marcelo Francisco Pompelli

Universidad de Córdoba, Montería, Colombia

Reviewer 2 Report

What type of experiment design did the authors use?

what does DCC in line 155 mean?

Needs more details about how the authors applied he drought stress.

Author Response

RESPONSE TO REVIEWER 2

Reviewer 2: What type of experiment design did the authors use?

Authors: The authors apologize for this lapse. The experiment design and statistical procedures was added in this version. Please verify it on lines 165-170.

Reviewer 2: what does DCC in line 155 mean?

Authors: This information is excluded (lines 109-110) and other section 2.6 Experimental design and statistical analyses was added in lines 165-170.

Reviewer 2: Needs more details about how the authors applied he drought stress

Authors: A better description of the treatments was presented between lines 104-114. The water stress was imposed by the total absence of irrigation. The authors thanks with your improvements.

To summarize, we have attempted to address all comments and have corrected all major and minor mistakes
outlined by the reviewer in the previous version our manuscript.

We truly believe that our manuscript is of suitable quality for publication in Sustainability.

Thank you for your attention.

Sincerely yours,

Dr. Marcelo Francisco Pompelli

Universidad de Córdoba, Montería, Colombia

Round 2

Reviewer 1 Report

The authors are acknowledged for the attempts made to improve the manuscript. While the manuscript has been improved, additional revisions are required. The gene expression results are not presented in a typical format (very often graphs are used to indicate up or down regulation of genes). It has been stated that there were no significant differences compared with the control but this is questioned given the data presented.

Attached are examples highlighted in yellow where sentence construction requires improvement - this provides an indication of the amount of improvements that are still required.

Author Response

The authors are grateful for your valuable contributions to improve this manuscript
